# Regenerative Potential of Injured Spinal Cord in the Light of Epigenetic Regulation and Modulation

**DOI:** 10.3390/cells12131694

**Published:** 2023-06-22

**Authors:** Samudra Gupta, Suman Dutta, Subhra Prakash Hui

**Affiliations:** 1S.N. Pradhan Centre for Neurosciences, University of Calcutta, 35, Ballygunge Circular Road, Kolkata 700019, India; samudra0073@gmail.com; 2Nuffield Department of Clinical Neurosciences, John Radcliffe Hospital, University of Oxford, Oxford OX3 9DU, UK; suman.dutta@ndcn.ox.ac.uk

**Keywords:** spinal cord injury, neurogenesis, axonal regrowth, histone modification, DNA methylation, extracellular vesicles

## Abstract

A spinal cord injury is a form of physical harm imposed on the spinal cord that causes disability and, in many cases, leads to permanent mammalian paralysis, which causes a disastrous global issue. Because of its non-regenerative aspect, restoring the spinal cord’s role remains one of the most daunting tasks. By comparison, the remarkable regenerative ability of some regeneration-competent species, such as some Urodeles (Axolotl), *Xenopus*, and some teleost fishes, enables maximum functional recovery, even after complete spinal cord transection. During the last two decades of intensive research, significant progress has been made in understanding both regenerative cells’ origins and the molecular signaling mechanisms underlying the regeneration and reconstruction of damaged spinal cords in regenerating organisms and mammals, respectively. Epigenetic control has gradually moved into the center stage of this research field, which has been helped by comprehensive work demonstrating that DNA methylation, histone modifications, and microRNAs are important for the regeneration of the spinal cord. In this review, we concentrate primarily on providing a comparison of the epigenetic mechanisms in spinal cord injuries between non-regenerating and regenerating species. In addition, we further discuss the epigenetic mediators that underlie the development of a regeneration-permissive environment following injury in regeneration-competent animals and how such mediators may be implicated in optimizing treatment outcomes for spinal cord injurie in higher-order mammals. Finally, we briefly discuss the role of extracellular vesicles (EVs) in the context of spinal cord injury and their potential as targets for therapeutic intervention.

## 1. Introduction

Spinal cord injury (SCI) in humans refers to the loss of sensory and motor functions above and below the injured area. The central nervous system, which includes the spinal cord, does not regenerate easily, leading to the death of nerve cells in the primary motor cortex [1,2]. SCI is a severe condition with a limited prognosis for recovery [3,4]. In developed countries, the annual occurrence of SCIs is approximately 25.5 cases per million people [5]. In adult mammals, the ability to regenerate after an SCI is hindered by inhibitors that prevent axons from growing and by the limited natural response for nerve cell development [3].

Unlike in the mammalian CNS, immature neural axons, peripheral nervous system (PNS) axons, and axons of certain non-mammalian animals such as teleost fishes, larval and adult urodeles (e.g., Axolotl), and anuran larvae (e.g., *Xenopus*) can recover from SCIs [6,7]. Although all of these models are useful for researching axon regeneration processes, here, we concentrate on the zebrafish (*Danio rerio*) due to the numerous recent studies that have explained some of the main molecules in this process.

The regulation of stage-specific gene expression in the nervous system during repair and regeneration involves not only transcriptional mechanisms, but also epigenetic modulation. Coined by Waddington in 1942, the term “epigenetics” emphasizes the complex interactions between genes, their products, and the environment, without any changes in the DNA’s primary sequence. Unlike classical genetics, which primarily focus on DNA sequence alterations, epigenetics encompass self-sustaining, reversible, and heritable changes in phenotypes that do not involve modifications in the DNA sequence [8].

In eukaryotic cells, gene expression is regulated by two main epigenetic mechanisms: DNA methylation and histone modification. These mechanisms modify the structure of DNA and its associated proteins to influence whether genes are turned on or off. Additionally, non-coding RNAs have recently emerged as important players in gene regulation. They help cells adjust their transcriptional responses to various signals in both normal and diseased conditions, and they also provide insights into the regenerative processes of cells [9].

The family of DNA methyltransferase (DNMT), which includes de novo (DNMT 3A, DNMT 3B) and maintenance methyltransferase (DNMT1), catalyzes DNA methylation [10]. DNA methylation blocks gene expression either by directly interacting with the transcription factor binding to DNA or by recruiting methyl-CpG binding domain (MBD) proteins that form complexes with histone deacetylase (HDAC) to turn chromatin into a repressive state [11,12]. 

Epigenetic regulation covers at least eight types of chromatin modifications that regulate gene expression without affecting the DNA sequences and modifying the histone tails [13]. DNA is packed into chromatin by wrapping four core histones (H2A, H2B, H3, and H4) around an octamer [14]. Histone tails are short N-terminal arms separated from the main structure and subjected to post-translational modifications, such as methylation, acetylation, and phosphorylation [9]. All of these, along with less frequent modifications, such as ubiquitination, sumoylation, ADP ribosylation, and deamination, change the structure of histone and, therefore, allow or prevent access to DNA [9].

Another of the most known modifications is the role of acetylation of lysine residues, a reversible mechanism catalyzed by either HAT or HDAC [3]. A member of the HAT family that adds an acetyl group transmits structural improvements as it reduces the contact between the negatively charged backbone of DNA and the positively charged tails of histones [3]. This decline in activity results in a less compacted nucleosome that is accessible to complexes of transcription factors [3]. Histone acetylation is also correlated with decreased transcription of chromosomes [15]. Conversely, HDAC eliminates the acetyl group, theoretically culminating in a general repression of gene transcription [3]. Histone methylation is another type of epigenetic control that has heritable long-term consequences but can also be reversed [3]. 

The key epigenetic processes and events are typically evolutionarily conserved and shared between zebrafish and mammals—specifically those that occur during the programming of germ cells [16]. Reversible acetylation/deacetylation, methylation/demethylation, and phosphorylation/dephosphorylation are involved in histone alteration [17]. Histone acetylation is carried out by histone acetylases (HATs), which display distinct trends of retention in the genome of zebrafish relative to rats, and they are variably influenced by events of gene duplication [18,19]. Histone methylation and demethylation are carried out by histone-lysine methylases (HKMTs) and lysine-specific demethylases (LSDs), respectively [20]. Ultimately, histone phosphorylation is controlled by histone–kinase and histone–phosphatases, which may influence the conformation and DNA association by either phosphorylating or dephosphorylating serine at the N-terminal of a histone [17]. DNA methylation in zebrafish is also performed by DNA-methyltransferases (DNMTs), which form 5-methylcytosine (5-mC) in the genomic CpG nucleotides by methylating cytosine, as in mammals [21]. 

MicroRNAs (miRNAs) are small non-coding RNAs that are approximately 22 nucleotides long and control post-transcriptional gene expression. When processed from longer stem–loop-like precursors, they are directed by base-pairing at the 3′UTR end to target mRNA sequences, leading to the breakdown of target mRNAs or translational repression [22]. The seed region is the essential region for miRNA, which binds Watson–Crick’s complementarity to the mRNA target site and consists of 2–8 nucleotides from the 5′ end of miRNA [23]. Some miRNAs are regulated in an epigenetic manner. At the same time, epigenetic-pathway-related molecules are regulated by a variety of miRNAs—particularly those in the Polycomb category of proteins: HDACs and DNMTs [24]. Altogether, post-transcriptional regulation by miRNAs and transcriptional regulation by epigenetic machinery collaborate to coordinate the whole gene expression profile and sustain the functionality of physiological cells [24].

To date, many studies regarding the epigenetic regulation behind SCI repair in mammals have been reported, but few studies have been carried out to find the epigenetic mechanism controlling the regenerative process after SCI in regeneration-competent animals (Axolotl, *Xenopus*), especially in zebrafish. This review presents and compares the intrinsic and extrinsic determinants of epigenetic regulation underlying both repair and regeneration in the injured spinal cord in mammals and zebrafish, respectively. A clearer understanding of the molecules and molecular mechanisms of epigenetic regulation that influence the extraordinary individualism in the CNS of mature zebrafish and other regeneration-competent animals may support the development and advancement of therapies for SCIs and their clinical translation in humans.

Extracellular vesicles (EVs) have become the focus of substantial research over the last few years. The membrane-delimited particles known as EVs are produced by living cells and released into the extracellular space. Exosomes, microvesicles, and apoptotic bodies are the three fundamental types of extracellular vesicles produced by cells. In the central nervous system, EVs regulate important physiological and pathological processes. During neurotrauma, such as spinal cord damage, EVs can act as pathologic regulators because they carry a range of bioactive cargo (such as nucleic acids, proteins, and lipids) that may be modified in response to external stimuli after injury. Here, we focused on the potential of EVs in the CNS in relation to SCI, in addition to the development of EV-based therapeutic improvements in preclinical models for such disorders.

## 2. Epigenetic Regulation in Mammals after Spinal Cord Injury 

### 2.1. Spinal Cord Injury: Primary and Secondary Injuries in Mammals

Spinal cord injury (SCI) may be defined as a physical harm imposed on the spinal cord that, either directly or indirectly, encompasses its four major functions (motor, sensory, autonomic, and reflex) [9]. The wide and complex array of motor, sensory, and autonomic problems that result from SCIs derive primarily from the severity of their lesions and the degree of disruption of the spinal cord after the secondary pathological development [25,26]. A basic understanding of the causes of pathophysiology in spinal cord injury is of prime importance in promoting the comprehension of pharmacological interventions [27].

Based on gross results, Bunge et al., (1993) provided a clear yet effective description separating human SCIs into four classes: (1) spinal cord contusion caused by an external injury, (2) impact with more transient compression by itself, (3) transection and laceration (most severe in short thoracic segments), and (4) distraction-forcible spinal cord stretching in the axial plane [27].

The severe SCI pathophysiology, which lays the foundation for long-term SCI deficits, includes both the primary and secondary phases of injury [9]. The primary injury results in immediate and direct mechanical obstruction of the activity of the spinal cord, mainly affecting the central gray matter while relatively sparing the white matter; this occurs mostly peripherally [4,28]. Primary injury includes the early occurrence of hemorrhage, resulting in hypoxia and ischemia, microhemorrhages, or edema near the injury site [25]. These cause interrupted nerve transmission, as neurons passing through the injury site are physically damaged and have reduced myelin thickness [25,29]. The gray matter is considered to be irreversibly damaged within the first hour of injury, whereas the white matter is irreversibly damaged within 72 h after the injury [25,29].

The most apparent mechanism is an energy loss induced by ischemia and poor cellular perfusion [30]. Ischemia can emerge promptly after a traumatic SCI, and if untreated, further deterioration can occur within the first 3 h and can last for at least 24 h [31]. Following SCI, several critical abnormalities are discovered, including hemorrhage, demyelination, edema, and cavity development with axonal and neuronal death, as well as a number of degenerative alterations in the nerve tissues, which can accelerate infarction [32,33]. Excitotoxicity, oxidative damage, and ischemia can be caused by elevated amounts of glutamate, whereas Ca^2+^-dependent nitric oxide synthesis can produce a secondary spinal cord injury [34]. Increased lipid peroxidation and free radical damage in the cell membrane, as well as additional injury signaling cascades in the injury site, can eventually cause neuronal death after secondary injury [26,35].

An initial primary immune response is required for primary damage clearance at the lesion site. However, a multifactorial process involving the recruitment of reactive astrocytes, microglia, macrophages, glial progenitors, fibroblasts, and Schwann cells during the secondary injury leads to the formation of persistent glial scarring [36,37,38]. This glial scar is often impenetrable and contains secreted and transmembrane molecules that inhibit the growth of the axon [39]. There may also be a gradual extension of the injury condition across more than one section (syringomyelia) over months or years, which often becomes catastrophic [39,40,41].

Following a spinal cord injury (SCI), supraspinal influence on the autonomic nervous system (ANS) is disrupted, which results in sympathetic blunting and parasympathetic dominance, neurogenic bradycardia, neurogenic orthostatic hypotension, adaptive myocardial atrophy, thermoregulatory dysfunction, neurogenic obstructive lung disease/airway hyper-reactivity, neurogenic bowel, neurogenic bladder, and neurogenic sexual dysfunction. For those who have a serious thoracic and cervical SCI, unrestricted and heightened reflex sympathetic outflow in response to unpleasant stimuli below the point of damage can also result in humiliating hyperhidrosis and a potentially fatal hypertensive crisis. Therefore, when providing treatment to this susceptible group, managing these autonomic abnormalities should be given top priority [42].

### 2.2. Mechanism of Injury Repair: Glial Scar Formation

In mammals, axonal regrowth after a spinal cord injury is hindered by the development of a glial scar that mainly consists of reactive astrocytes and proteoglycans, which tend to perform important defensive functions while stabilizing the tissue of the fragile CNS [40,43]. Injured axons on the rostral or cell-body side of the injury site withdraw in a process called the dieback cycle and develop dystrophic end bulbs that maintain the capacity to generate new growth cones unless the proper environment is provided [44]. On the caudal side of the injury, the sectioned axons separated from their neuronal cell bodies undergo Wallerian degeneration within 24–48 h, leading to the loss of all caudal motor and sensory features at the injury site [45,46,47].

Furthermore, adult CNS neurons do not express many of the genes known to boost axon regeneration after damage. Manipulation of the expression of regeneration-associated genes (RAGs) is one method of promoting CNS regeneration. Recent research has indicated that inhibiting factors, such as PTEN, Klf4, and SOCS3, as well as the overexpression of genes activated during PNS regeneration, such as CREB, cJun, and Klf7, can promote axon development in the adult CNS [48,49,50,51].

SCI causes tissue ischemia as a result of vascular stress and edema [25]. The injured cells produce ATP, which activates a number of purinergic receptors expressed by astrocytes, microglia, and oligodendrocyte precursor cells (OPCs) [52]. Furthermore, astrocyte activation can be induced shortly after SCI and exacerbated by a variety of inflammatory factors (IL-1b, IL-6, IFN-γ, TNF, etc.) [53].

Within a 7–10-day period of glial scar development, certain reactive astrocytes rapidly proliferate and densely occupy the region surrounding the lesion’s center [54]. The proliferation of these astrocytes is fastest 3–5 days after damage, declines 7 days later, and almost stops 14 days later. These astrocytes begin to grow elongated processes parallel to the injury’s epicenter throughout this procedure. They progressively grow thick, eventually forming a thin restricting boundary (the glial scar) around the injury’s epicenter. As a result, these astrocytes might receive signals from the and undergo phenotypic transformation into scar-forming astrocytes [55].

Around 2 weeks after an injury, the rapid proliferation of reactive astrocytes stops considerably. Scar-forming astrocytes complete their phenotypic transformation and no longer orient their processes parallel to the injury site, but rather become increasingly parallel and overlap with one another. The SCI lesion begins to stabilize at this stage, and a protracted chronic period of regeneration commences. The STAT3 pathway, a key modulator of astrocyte activation, is also involved in glial scar maturation [56].

Damage to the spinal cord triggers the degeneration of nerves and glial cells around the lesion’s locations [13]. Endogenous neural stem cells around the central canal cannot successfully lead to functional healing after an SCI in the spinal cord as they would under normal conditions; the activity of such populations of neural stem cells is quite limited [13]. However, few cells from the neural stem or precursor cell (NS/PC) reservoirs in the central nervous system (CNS) may be replenishing a certain number of reduced neurons [57,58]. Furthermore, although a limited number of neurons survive cell death, adult mammals cannot fully regenerate a damaged neuronal network in the CNS once it has been damaged [58,59]. Therefore, the only remaining option for regenerating injured areas is the use of the surviving neurons [58].

In this case, it is extremely important to generate correct lineages of cells according to the degree and form of the SCI for effective recovery from the injury [13,60]. The sequential generation of neurons accompanied by glial cells controls the differentiation of NS/PCs during CNS development, which is regulated by endogenous epigenetic mechanisms [61,62].

### 2.3. Epigenetic Regulation during Injury Repair

#### 2.3.1. Epigenetic Regulation of Specific Cell Types and Secondary Damage Processes

Following an SCI, the second extended phase of subacute inflammatory responses is initiated by a number of rapid alterations in inflammatory signaling that are closely related to the magnitude and nature of the impact [63]. Two primary macrophage phenotypes (M1 and M2) have been identified as having distinct functions that change the balance of neurotoxicity and regeneration in damaged mouse spinal cords, and these roles are critical for both the acute and persistent SCI response [64]. The “classical” pro-inflammatory macrophage subtype is M1, which is activated by interferon-gamma (IFN-γ) and generates significant amounts of pro-inflammatory cytokines. Following an SCI, M1 and M2 are both engaged, but M2 decreases and M1 continues after 7 days, leading to a protracted inflammatory response. HDAC inhibition might improve healing while minimizing inflammatory damage, as the epigenetic state appears to be important in determining the phenotypes of macrophages and T-cells throughout their responses to injury [65]. Histone acetylation in macrophages can be increased by IFN (from active T-cells) and Toll-like receptor activation, which also supports the M1 phenotype. In response to cytokines such as IL-4, HDAC3 acts as a direct regulator of transcription factors that promote M1 in the alternate activation of M1 and M2 states. An increase in M2 macrophages, which provide a pro-repair, anti-inflammatory milieu, is made possible when HDAC3 is eliminated [66]. Jmjd3m, a demethylase necessary for M2 polarization in response to concurrent activation by NF-κB and Toll-like receptors, is also involved in this balance of responses, together with PPAR [67].

Microglia also experience a unique set of morphological and metabolic alterations that can directly cause both short-term and long-term neuropathic pain after an SCI [68]. Microglial cathepsin S is responsible for immediate pain after damage through the activation of the p38 MAPK pathway and may be blocked by VPA (valproic acid), a broad-spectrum HDACi. Therefore, the intensity and scope of the immune response after an SCI may be affected by alterations in histone deacetylation [9].

Glial fibrillary acidic protein (GFAP) and cyclooxygenase (COX)-2 are elevated in astrocytes throughout reactive gliosis, while iNOS (inducible nitric oxide synthase), NO (nitric oxide), and IL-6 are upregulated in both astrocytes and microglia [9]. In various injury environments, HDAC inhibition can control each of these genes—typically, to reduce inflammation. Additionally, some glycosaminoglycans, such as heparin, can function as HAT inhibitors and prevent possible increases in acetylation levels in other cells by upregulating glycosaminoglycans, such as chondroitin sulfate proteoglycans (CSPGs) [9]. 

In a hypothetical repair scenario following SCI, oligodendrocytes perform two competing functions. Their myelin components, in particular, have a significant inhibitory effect on axonal regrowth [69]. Inhibiting oligodendrocyte differentiation while removing myelin trash probably promotes spontaneous sprouting and regeneration in spinal circuits [70]. Determining the time windows after an SCI during which to first suppress OPC differentiation during a period of neurite outgrowth and then permit HDAC activity to resume remyelination will be a crucial part of understanding the action of HDACs inside OPCs.

#### 2.3.2. Histone Modification during Spinal Cord Repair

Histone acetylation is regulated by combining the activity of histone acetyltransferases (HATs) and histone deacetylases (HDACs). HATs improve histone hydrophobicity through the transition of the active acetyl group from coenzyme A to histone lysine residues. The acetylated chromatin transforms into a less condensed form to facilitate the expression of genes. In comparison, during transcription, HDACs exert a silencing impact on genes [71]. A lower degree of acetylation in histones is desirable for developed neurons to preserve their structural and functional stability [72]. This stability, however, normally reduces the repair capabilities of the neurons. Recent studies have shown that axons’ low regenerative efficiency is possibly triggered by H4 hypoacetylation, which prevents post-injury axon reprogramming [72]. 

The HDAC family is subdivided into four categories based on structural differences and response mechanisms [73]. The catalytic activity of Class I, II, and IV HDACs is zinc-dependent, while the deacetylase activity of Class III HDACs (Sirtuin 1–7) relies on the co-factor NAD+ [73]. Class I HDACs (HDAC1, 2, 3, and 8) are primarily found within the nucleus [74]. In comparison, Class IIa (HDAC4, 5, 7, and 9) and Class IIb (HDAC6 and 10) shuttle HDACs between the cytoplasm and nucleus [74]. Class IV comprises only one member, HDAC11, which is extensively expressed but remains elusive in its function. In transcriptional complexes, HDACs usually act as epigenetic co-repressors that are recruited to particular genomic loci [75]. Nonetheless, HDAC substrates are not limited to histones, as many isoforms are clustered in the cytoplasm and can catalyze non-histone protein deacetylation [75].

#### 2.3.3. Histone Modification during Axon Regeneration

When neurons in the peripheral nervous system (PNS) are damaged, they can enter a growth state and activate a range of regeneration-associated genes (RAGs) that promote regeneration. However, in the central nervous system (CNS), mature neurons undergo long-lasting changes in gene activity, which poses a challenge for axon regeneration. Unlike embryonic neurons, which naturally have a high potential for axon growth, these modifications in mature CNS neurons are not easily reversed after damage [75,76]. For example, in fetal and newborn mice, actively growing corticospinal axons express SOX11, a key transcription factor required during axon regeneration in peripheral nerves; however, after injury, mature neurons do not express SOX11 [77]. A study has found that throughout later development and neuronal maturation, genes associated with neuronal growth are downregulated [78]. They found that this downregulation is related to modifications in chromatin accessibility, whereas upregulation is associated with chromatin decondensation [78]. A decrease in chromatin accessibility correlates with a decrease in the expression of genes such as *Sox11* and *Klf7* (another transcription factor that promotes axon regeneration) when neurons lose their intrinsic potential for axon growth [78].

Prior to the activation of axon-growth-associated genes in a peripheral nerve, when axon regeneration is effective, chromatin remodeling genes and histone acetyltransferases exhibit enhanced expression [79]. Additionally, injury to the peripheral nerve causes nuclear export of HDAC5, which is necessary for the activation of genes linked to regeneration [80].

Sensory neurons in the dorsal root ganglia (DRG) implant a central branch into the spinal cord, which is refractory for axotomy regeneration [81]. Injury to their peripheral branch can activate a retrograde transmission of injury signals, leading to a nuclear response that improves not only the peripheral but also the axonal growth potential of the central branch [81]. The critical role of histone-modifying enzymes in the epigenetic reprogramming of axon regeneration in DRG neurons has been revealed both recently and in earlier studies. Studies have found that levels of both global and RAG-specific histone 4 acetylations (AcH4) in mature DRG neurons are low [82]. Pharmacologically increasing the rates of histone acetylation by administering HDAC inhibitors such as trichostatin A (TSA), a large inhibitor of Class I and II HDACs, or MS-275, a more selective HDAC inhibitor, may trigger several RAGs and promote sensory axon regeneration after dorsal column transection [82]. Puttagunta and colleagues demonstrated an increase in histone 3 lysine 9 acetylations (H3K9ac), a marker of actively transcribed genes, and a decrease in H3K9 methylation (H3K9me2), a repressive marker, among promoters of GAP43, Galanine, and brain-derived neurotrophic factor (Bdnf) after studying residue-specific histone acetylation in DRG neurons after peripheral axotomy [83]. An H3K9ac-specific acetyltransferase, p300/CBP-associated factor (P/CAF), is enhanced by the promoters of these RAGs and is dependent on retrograde signaling through the activation of the pathway of extracellular signal-regulated kinase (ERK) [83]. P/CAF overexpression increases post-SCI regeneration of ascending sensory neurons [83]. 

These acetylation findings are in accordance with data showing that extensive accessibility of chromatin around cell-fate transitions enhances regeneration, which is likely due to reciprocal changes in HAT/HDAC activity.

#### 2.3.4. Role of DNA Methylation in Injury Repair

DNA methylation has also been investigated due to its potential involvement in the regulation of regeneration-associated genes in response to injuries in peripheral nerves, in addition to the evidence for histone acetylation and regeneration. There are core and peripheral branches in the sensory neurons of the dorsal root ganglion. DNA methylation is upregulated after peripheral branch axotomy, which effectively promotes axon regeneration [84]. However, DNA methylation is not upregulated after central branch axotomy, which does not enhance regeneration [84]. Increases in the methylation of axon development and myelination genes are observed when Tet3 is elevated after injury [84]. There is a significantly greater expression of regeneration-associated genes in peripheral axotomy compared to central axotomy, and these are methylated within 24 h after injury [85].

#### 2.3.5. miRNA Regulation of Injury Repair

MicroRNAs (miRNAs) are a class of tiny non-coding RNAs that are involved through post-transcription control in several biological processes [86]. Recent studies have shown that Dicer-dependent miRNAs are engaged in controlling post-injury axon regeneration. There was less regeneration of the sensory nerves after nerve damage in Dicer-knockout mice when evaluated with different criteria [87]. In a rodent SCI model, the scope of miRNA expression changed significantly. Many modulated miRNAs control RAGs that handle regeneration through axons [88].

Subsequently, miRNAs have turned out to be essential intrinsic epigenetic regulators that regulate post-injury regeneration in either central or peripheral axons. MiR-132 functions as a supportive regulator, facilitating axon extension during growth by blocking its downstream target, RasGTPase activator Rasa1 [89]. Interestingly, miR-132 is one of the miRNAs that is upregulated after ischemic spinal cord injury [90].

In an SCI model for rat ischemia–reperfusion, miR-210 was more than downregulated by more than two fold [91]. In another experiment, Hu et al., (2016) observed that repression of endogenous miR-210 in dorsal root ganglions hindered the regeneration of both in vitro and in vivo axons and that inactivation of ephrin-A3 (EFNA3) effectively retrieved axons from their failure to regenerate [92].

Another study indicated that miR-210 overexpression facilitates Wnt-pathway-dependent anti-apoptosis and angiogenesis, thus enhancing the regeneration of post-SCI neurological activity [93]. These findings indicate that the downregulation caused by miR-210 after an SCI is possibly detrimental to post-injury axon regeneration, and the control of miR-210 expression is considered to be a possible goal for future clinical SCI studies. Additionally, the author’s group observed that miR-26a expression facilitates mammalian sensory axon regeneration by inhibiting Smad1 inactivation mediated by glycogen synthase kinase 3β (GSK3β) [94]. Another group of researchers found that miR-409 expression in mice with SCI was downregulated, while overexpressed miR-409 increased the gripping power of mice with SCI to some degree by specifically targeting ZNF366 [95]. Jiang et al., also stated that miRNA 21 can suppress the progression of glial scars and boost spinal cord repair by suppressing the expression of the pro-apoptotic Pdcd4 gene in the secondary phase of SCI [94]. In fact, miRNAs and other epigenetic modifiers create an epigenetic regulatory feedback loop, such as the miR-138/SIRT1 negative feedback loop in axon regeneration control, to collectively regulate gene expression [96]. While the molecular processes influencing axon regeneration and the pathogenesis of SCIs are yet to be investigated, these findings sufficiently show the close association between epigenetic influences and axon regeneration within the CNS [72]. Consequently, a beneficial orchestration of epigenetic factors and regulatory networks is expected to transform mature sleeping neurons into a capable regenerative state, which may help in establishing potential new therapeutic strategies for SCI therapy [72].

## 3. Epigenetic Regulation of Regeneration-Competent Animals after Spinal Cord Injury

### 3.1. Axolotl

Some vertebrates have retained an exceptional capacity to rebuild a completely functioning spinal cord after amputation or damage to the tail [97,98]. In the Mexican axolotl salamander, this process includes the development of terminal vesicle-like structures on both the rostral and caudal ends of the wounded neural tubes of terminal vesicle-like structures [99]. Cells within ~500 microns of the injury site continue to separate and move to cover the missing or injured part of the neural tube [100]. Various genes that are up- or downregulated in glial cells at the damaged ends of the spinal cord have been identified; these cells subsequently proliferate to heal the lesion [6]. No proof of glial scar development is evident in axolotl salamanders, as determined by measuring the pre-injury and post-injury levels of GFAP, vimentin, chondroitin sulfate proteoglycans (CSPGs), or collagen [7]. Axons regenerate from the initial lesion site, resulting in caudal motor and sensory controls returning after injury [101]. How functional recuperation occurs in axolotls is still unclear. It is unknown if damaged axons regrow and reconnect to the same sites or if new neurons are formed, thus generating new neuronal associations, as not many case reports have been recorded on the epigenetic modulation of regeneration in axolotl after SCI.

One group of researchers described a population of miRNAs that are maintained between the Mexican axolotl salamander (*Ambystoma mexicanum*) and mammals that exhibit pronounced cross-species variations in patterns of control after spinal cord injury [102]. They observed that specific post-injury rates of one of these miRNAs (miR-125b) are important for functional recovery and guide proper axon regeneration via the lesion site in a process involving a direct downstream target, *Sema4D*, a member of the *semaphorin* gene class in axolotls (Figure 1A) [102]. Another study showed that HDAC activity is necessary for controlling the initial transcriptional response to injury and regeneration in axolotl at the time of tail amputation [103]. It was also observed in another study that miR-196 functions as an important regulator of tail regeneration in axolotl that acts upstream of the main patterning events within the spinal cord based on *bmp4* and *pax7* [104]. miR-196 acts directly on *pax7* to downregulate the rates of Pax7 protein in cells in the 500 μm region anterior to the amputation path, which, in turn, serves as a cue to the cells to increase their proliferation and to move to form a new ependymal channel [104]. In a separate axolotl study, miR-200b was found to enhance the production of c-Jun after SCI, thus effectively inhibiting the formation of the typical AP-1^cFos/cJun^ complex. This inhibition, in turn, prevented the surrounding glial cells from upregulating the expression of AP-1^cFos/JunB^, a factor responsible for initiating a cascade of events leading to glial cell proliferation and functional recovery at the injury site (Table 1) (Figure 1A) [105].

### 3.2. Xenopus

African clawed frog tadpoles, *Xenopus laevis*, have the potential to rapidly recover their tails after amputation. Knowledge of the tail regeneration pathways will contribute to new ideas for fostering the biomedical regeneration of non-regenerative tissues. Although chromatin remodeling is understood to be important for stem cell pluripotency, its function in the regeneration of complex organs in vivo remains almost entirely uncharacterized. Histone deacetylase (HDAC) activity was found to be essential for the early stages of tail regeneration and a novel role for HDAC activity was established during the early stages of tail regeneration in *Xenopus* [107,112]. A Class I HDAC, HDAC1, and other unspecified HDACs are strongly expressed during endogenous regeneration [107]. Pharmacological blockage of HDACs by utilizing Trichostatin A (TSA) in amputated tails improved histone acetylation rates [107]. Treatment with TSA or other HDAC inhibitors, such as valproic acid, explicitly prevented regeneration. Inhibition of HDAC function resulted in an aberrant expression of *notch1* and *bmp2*, two genes considered to be important for tail regeneration [107].

HDAC action is required to properly establish two inflammatory genes, *mpox* and *spib*, whose pattern of expression produces an inflammatory response that is a key player during *Xenopus* tail regeneration [108]. Suzuki et al., also reported that *Xenopus* tail amputation induced ROS signaling, which was preceded by a rise in the level of H2K9ac, which contributed to the reactivation of regeneration genes (*shh, fgf20*) in notochord regeneration and represented a signaling site for orchestrating tail formation (Figure 1B) (Table 1) [106].

### 3.3. Zebrafish

#### 3.3.1. Spinal Cord Injury Response in Zebrafish

Unlike in mammals, in zebrafish, spinal cord injury responses tend to be quite different, and they result in injury healing and functional recovery. Several cellular responses that are distinct from those in mammalian SCI are the following: (a) the development of a very brief inflammatory reaction mediated by numerous gene sets [113,114]; (b) the involvement of macrophages at the wound site, which is possibly involved in the clearance of myelin debris [115,116,117] and the upregulation of anti-inflammatory M2-type macrophage-related molecules, unlike in mammals, where pro-inflammatory macrophages are found at the lesion site and could be responsible for sustained dieback of damaged axons [64,118]; (c) rather limited cell loss due to necrosis and apoptosis following injury (although apoptotic cell death is normal in both mammalian and zebrafish SCI, the degree and magnitude of cell death vary in their spatiotemporal patterns and include upregulation of specific molecular sets relative to mammalian SCIs) [119]; (d) generation of permissive axonal regrowth conditions [119]; (e) proliferative response and pervasive neurogenesis [119]. Endogenous ventricular progenitor cells known as ependymo-radial glial cells (ERGs) accomplish restorative neurogenesis. The ERGs remain dormant under homeostatic settings but are stimulated to proliferate and differentiate.

#### 3.3.2. Epigenetic Regulation of Spinal Cord Regeneration in Zebrafish after Injury

Zebrafish organs are subject to ongoing research because of their high regenerative capacity, including in the tail fin, heart, pancreas, kidney, retina, brain, and spinal cord [120]. Significant progress has been made in uncovering the cellular and molecular processes that drive organ regeneration, leading to a wide range of novel findings. Several observations elaborating on the epigenetic control behind the regeneration of different organs (e.g., fin, brain, and retina) in zebrafish have been reported, but the actual epigenetic mechanism underlying spinal cord regeneration after injury remains unclear. Here, we present a brief review of the epigenetic mechanisms underlying SCI regeneration in zebrafish based on a few observations that have been reported (Table 1).

Recent research on adult zebrafish has shown that miR-133b’s exogenous overexpression not only facilitates post-SCI axon regeneration, but also significantly increases motor function recovery by regulating the RhoA signaling pathway (Figure 2) [109]. miR-133b is important for locomotive recovery and axon regeneration in adult zebrafish [109] and can enhance locomotive recovery after spinal cord injuries in mice [121]. Nevertheless, the expression of miR-133b in a single Mauthner cell inhibited axon regeneration in a model utilizing two-photon axotomy in zebrafish embryos, and its inhibition promoted axon outgrowth via the modulation of *tppp3*, which may stimulate axon regeneration [110]. Such conflicting findings on the role of miR-133b could be related to the use of multiple injury models, which needs to be addressed in future studies.

In a recent study, the role of Hdac1 in the ERG during spinal cord regeneration in zebrafish was investigated [111]. After spinal cord transection in larval zebrafish, the reduced expression of Hdac1 in ERGs by a potential Hdac1 inhibitor decreased the numbers of both newborn motor neurons and total newborn neurons (Figure 1C) [111]. This decrease was observed to be caused by a reduction in the ERG proliferation driven by a lesion. Hdac1 is, therefore, an advantageous regulator of regeneration in the spinal cord of zebrafish. The quantity of newly developed motor neurons after spinal cord injury was unaffected by increased deacetylation caused by Hdac1 overexpression or global Hat inhibition in the ERG [111]. 

## 4. Extracellular Vesicles (EVs) in the Epigenetic Regulation Process

Extracellular vesicles (EVs) are small membrane-bound nanoparticles that are secreted by cells into the extracellular environment and contain various cell- and cell-state-specific biomolecules, such as proteins, lipids, and nucleic acids, including mRNAs and miRNAs [122,123,124]. EVs have been shown to play a critical role in intercellular communication [125,126], including a role in the epigenetic regulation of gene expression by transporting their cargo biomolecules from one cell to another, thereby influencing the recipient cell’s gene expression [127]. One way in which EVs are involved in epigenetic regulation is through the transfer of miRNAs, which are small, non-coding RNA molecules that can regulate gene expression by targeting specific mRNA molecules for degradation or repression of translation. miRNAs are known to be present in EVs and can be taken up by recipient cells, where they can then modulate gene expression in the recipient cells [128]. This process may contribute to the transfer of epigenetic information between cells and potentially alter the epigenome of the recipient cells [129,130]. In addition to miRNAs, EVs may also contain other biomolecules that can alter gene expression, such as DNA methyltransferases and histone-modifying enzymes (Table 2). These enzymes can modify the epigenomes of recipient cells by adding or removing epigenetic marks, such as through DNA methylation or histone modifications.

In the context of spinal cord injury, EVs have been reported to be involved in epigenetic regulation during injury-induced changes [131]. EVs released by injured cells can carry cargo with chemical modifications, and the transfer of these materials to other cells can lead to changes in gene expression in the recipient cells. Specifically, EVs have been found to carry microRNAs, which can regulate the expression of genes involved in spinal cord repair and regeneration (Figure 3A) [132]. For example, studies have reported that EVs isolated from the spinal cords of mice with SCIs contain miRNAs that regulate gene expression by binding to specific mRNA targets and inhibiting their translation. One example of this is the microRNA miR-21, which has been found to be upregulated in EVs released by injured spinal cord tissue [133]. The overexpression of miR-21 may also prevent neuroinflammation, enhance blood–spinal cord barrier performance, control angiogenesis, and inhibit the growth of glial scar tissue. However, on the contrary, miR-21 has been shown to inhibit the expression of genes involved in axon growth and regeneration, thus potentially contributing to the inability of spinal cord tissue to repair itself after injury [134]. 

**Table 2 cells-12-01694-t002:** Summary of the biomolecules identified in EVs, which are potentially involved in epigenetic regulation, along with their functions.

Molecules	Functions	Examples in Extracellular Vesicles	References
**microRNAs (miRNAs)**	Regulate gene expression by binding to messenger RNA (mRNA) molecules, leading to mRNA degradation or inhibition of translation. They can be packaged into EVs and transferred between cells, thus influencing epigenetic processes.	**miR-21:** Regulates the differentiation and death of neurons in patients with SCIs	[135,136]
		**miR-30a, miR-145, miR-155, and miR-216:** Higher expression in SCI patients compared to uninjured adults	[137]
		**miR-126:** Stimulates angiogenesis and promotes regeneration of neurons while decreasing cell death in rats with SCIs	[138]
		**miR-29b:** Heals injured spinal cords in rats	[139]
		**miR-133:** Regenerates axons, preserves neurons	[140]
**Long non-coding RNAs (lncRNAs)**	Regulate gene expression by interacting with DNA, RNA, and proteins. Some lncRNAs have been detected in EVs, and their transfer can potentially affect epigenetic regulation and gene expression in recipient cells.	**lncGm3749:** High expression in EVs under hypoxic conditions, effective in repairing SCI by suppressing inflammatory mediators.	[141]
		**lncPTENP1:** Helps in recovery from SCI by regulating the expression of miR-21 and miR-19b	[142]
		**lncTCTN2:** Improved functional recovery after SCI	[143]
**Circular RNAs (circRNAs)**	Some circRNAs have been identified in EVs, and they can potentially act as carriers for miRNAs. Their transfer through EVs may influence gene expression and epigenetic processes in recipient cells.	**circZFHX3:** Inhibits LPS-induced BV-2 cell injury, suggesting a potential therapeutic strategy for the treatment of SCI.	[144]
		**CircRNA CDR1as (ciRS-7)** has been detected in EVs and shown to act as a sponge for miR-7, thus regulating gene expression.	[145]
**Histones**	Though histones are primarily found within cells, recent studies have identified histones and histone-associated complexes in EVs.	Histones H3 and H4, along with the associated proteins, have been found in EVs, indicating their potential involvement in epigenetic regulation and intercellular communication.	[146]
**DNA Methylation**	While EVs have been found to contain DNA, the specific presence of methylated DNA in EVs and its role in intercellular communication and epigenetic regulation are areas of ongoing research.	DNA methyltransferases (DNMTs) and mRNA have been detected in EVs, suggesting their potential transfer and influence on gene expression in recipient cells.	[129]

EVs contain a diverse range of protein and nucleic acid cargos, but their regenerative effects primarily stem from the transfer of specific proteins and miRNAs [147]. Several miRNAs found within EVs, including miR-133b, miR-1000, miR-34a, miR-219, and miR-21, have been identified as key regulators of neuroprotection, synaptic glutamate release [148], neural plasticity promotion [149], and myelination enhancement [150]. These findings highlight the critical role of specific miRNAs carried by EVs in mediating various regenerative processes. In addition to miRNA, EVs released from injured spinal cord tissue also contain DNA methylation patterns that differ from those found in healthy tissue. These epigenetic alterations may contribute to the long-term changes in gene expression observed after SCI.

A recent investigation conducted by Liu et al., revealed promising findings regarding the potential of EVs derived from bone marrow stem cells (BMSCs) for facilitating the formation of blood vessels, inhibiting the formation of glial scars, reducing neuronal cell death, alleviating inflammatory reactions, and supporting the regeneration of axons [151]. Consequently, these effects contribute to the restoration of functional behaviors following acute SCI. Notably, one plausible mechanism underlying these beneficial outcomes could be the suppression of A1 neurotoxic reactive astrocytes’ activation. Collectively, their findings suggest that utilizing EVs obtained from BMSCs holds significant promise as a viable strategy for treating SCIs [151]. It is to be noted that most preclinical investigations have centered on the transfer of specific proteins or miRNAs via EVs, thus demonstrating their potential for promoting neuro-regeneration. However, as suggested in [152], it is likely that the therapeutic effects and advantageous outcomes of EVs arise from the transmission of a diverse range of molecules encompassing signaling lipids, growth factors, miRNAs, and more, rather than relying on the influence of a solitary molecule.

Spinal cord injury in zebrafish can cause the release of EVs from damaged cells. In addition to their potential use as biomarkers, EVs released after SCI in zebrafish may also play a role in the healing process, though at present, the available information on this is very limited. One recent finding indicates that during the process of regenerating the caudal fin in zebrafish, cells known as blastemal cells utilize EVs as a means of communicating with other cells [153]. These EVs have been shown to contain proteins that are involved in inflammation and cell death, suggesting that they may contribute to the damage during spinal cord injury. 

## 5. Discussion and Therapeutic Approaches

Epigenetics is not a so-called ‘new era’ in the field of regenerative research, as many groups of scientists have been engaged in this field for decades. However, the main epigenetic mechanisms behind the processes of repair and regeneration in both non-regenerating and regenerating animals, respectively, remain unclear. This review provides a clear concept of spinal cord injury in both non-regenerating and regenerating animals and the epigenetic bases of repair and regeneration in a comparative way.

After SCI, restoration of paralysis in mammals remains one of the most challenging tasks in all neuroscientific research. Given major advances in early diagnostic and surgical control of SCI, along with a significantly enhanced knowledge of the pathophysiology of SCI, no appropriate therapies for enhancing the neurological conditions following SCI remain in operation because of the non-regenerative nature of the mammalian CNS. Recent research has demonstrated that epigenetic changes and related controls are implicated in the main facets of SCI recovery, including axon regeneration, glial activation, inflammatory reaction, and endogenous NSC reprogramming. Both breakthroughs are propitious candidates for SCI studies and attractive goals for clinical treatment of SCI. Such recent findings will set up accurate biomarkers of epigenetic networks for predicting the prognosis and clinical assessment of SCI. The results for epigenetic modifications caused by SCI may be better clarified for therapeutic applications for the development and optimization of bioinformatic repositories.

Several researchers have tried to restore damaged neuronal circuits through cell transplantation with the introduction of embryonic stem cells and induced pluripotent stem cells (iPSCs), and the transplantation of NS cells into a damaged CNS has been shown to be an effective treatment [154,155,156,157,158,159,160,161]. In order to improve functional recovery, epigenetic manipulation to control NS/PCs and their microenvironments at damaged sites may facilitate neuronal differentiation of NS/PCs and axon elongation [13]. In the context of nerve cell regeneration, echogenic progenitors are specialized cells that are frequently introduced or transplanted into a specific location in the body with the aim of generating new nerve cells and facilitating tissue repair. Understanding how epigenetic regulation affects these echogenic progenitor cells can provide insights into their differentiation, maturation, and functional integration into the nervous system. It may also have implications for optimizing their therapeutic use for various neurological conditions and diseases [162,163].

The finding of dramatically altered miRNA expression following SCI not only exposes the different pathways behind this traumatic progression, but also provides possibilities for future therapeutic approaches. This analysis summarizes the results of recent studies—mainly those involving post-SCI expression analysis—concerning the possible roles of these small non-coding RNA molecules in several post-injury processes, such as inflammation, apoptosis, glial scar development, and axonal regeneration. MiRNAs have enormous potential for becoming a new class of therapeutic medicines, but possible issues, such as high-dose-associated side effects and toxicity when administered in vivo, as well as unexpected off-target effects of specific miRNAs, exist. miRs are pioneers in translational medicine research pertaining to epigenetics because of their modest size and conserved nucleotide sequence. Numerous clinical trials using liposomes and antisense nucleotides have been started as a result of advances in chemical synthesis and drug delivery techniques. For instance, PF-655 by Quark and Pfizer, which promotes RTP801 gene expression, and SPC3649, an anti-HCV medication by Santaris, are based on the miR-122 antisense nucleotide. These medications’ therapeutic efficacy, tolerability, and safety have all been extensively established in prior clinical studies. However, there are still a few issues that need to be resolved, such as the hybridization linked to off-target effects and delivery-related issues [164,165,166].

Animals, including salamanders and zebrafish, share certain aspects of their physiology with humans, but separate molecular mechanisms behind injury response have emerged. It is currently unclear if the capacity to achieve adult regeneration is an inherited feature that has been lost through mammalian evolution or whether such abilities have evolved spontaneously through local evolution. Nonetheless, exploiting our understanding of animal regeneration is hoped to improve our existing clinical therapeutic strategies by offering information about how to treat mammalian SCI.

Recently, EVs have gained significant attention as potential therapeutic vehicles for delivering cargo to target cells, which is mainly due to their ability to penetrate the blood–brain barrier (BBB) [167,168] However, so far, research investigating the potential therapeutic value of EVs in spinal cord injury is scarce. One possible use of EVs in the treatment of spinal cord injury is to deliver therapeutic agents, such as growth factors or stem cells, directly to the site of the injury (Figure 3B) [169]. This approach may be able to promote the regeneration of damaged nerves and improve functional recovery. For example, EVs isolated from stem cells have been shown to promote the regeneration of damaged spinal cord tissue in animal models. However, it is worth noting that many studies investigating the regenerative potential of EVs derived from stem cells often prioritized evaluating the overall therapeutic efficacy, rather than providing comprehensive insights into the specific physiological or biochemical changes that occur due to the cargo carried by EVs [170]. The other potential use of EVs in SCI is as a diagnostic tool. EVs released by cells at the site of the injury may contain biomarkers that can be used to assess the severity of the injury and monitor the response to the treatment. Additionally, EVs secreted from neighboring healthy cells may also have an immune-modulatory effect, helping to reduce inflammation and promote tissue repair. In preclinical studies, EVs from immune cells have been shown to reduce inflammation and improve functional recovery in animals with SCIs.

MicroRNAs carried by EVs have been shown to regulate key pathways involved in neuroplasticity, axon guidance, and remyelination, thus providing a potential avenue for promoting spinal cord regeneration. Additionally, EVs can transfer long non-coding RNAs, which are emerging as important regulators of gene expression and have been implicated in various neurodevelopmental and regenerative processes. Furthermore, EVs can also carry DNA methylation modifiers, allowing for the regulation of DNA methylation patterns that control gene expression. Overall, the role of EVs in the epigenetic regulation of gene expression after SCI is complex and multifaceted and is an active area of research. Though it is clear that these small vesicles play a crucial role in the injury response, further studies are needed to fully understand the mechanisms by which EVs contribute to the regulation of gene expression in this context, and a better understanding of these mechanisms may lead to the development of new therapeutic strategies for SCI.

Despite the exciting prospects of EV-mediated epigenetic regulation, several challenges must be addressed to optimize their efficacy and safety. Standardization of EV isolation and characterization protocols is crucial to ensure the reproducibility and comparability of experimental results. Furthermore, improving methods for efficient loading of specific epigenetic cargo into EVs and enhancing their stability during storage and transport are essential considerations. Rigorous preclinical studies utilizing appropriate animal models and relevant outcome measures will be crucial for evaluating the therapeutic potential of EV-based epigenetic regulation. Finally, the translation of EV-based therapies for spinal cord regeneration to clinical trials necessitates addressing regulatory and manufacturing challenges, as well as ensuring the development of safe and scalable manufacturing processes.

## Figures and Tables

**Figure 1 cells-12-01694-f001:**
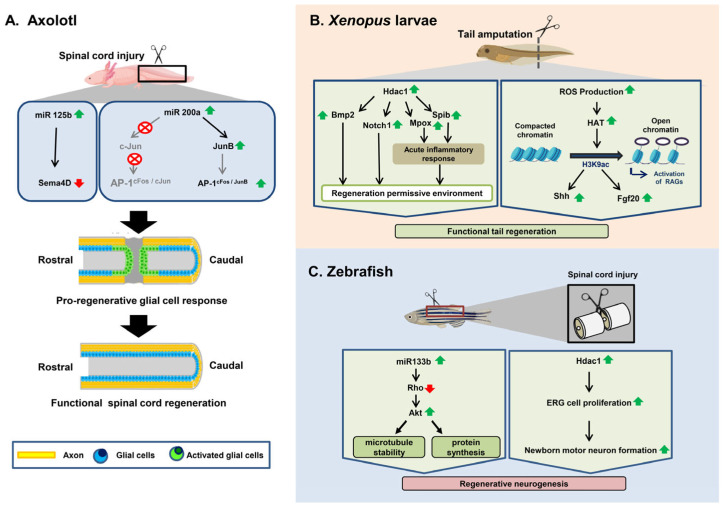
Comparative view of different epigenetic regulations underlying spinal cord regeneration in the axolotl (**A**) and zebrafish (**C**) and tail regeneration in Xenopus larvae (**B**).Red arrows and green arrows indicate downregulation and upregulation of genes and other factors, respectively.

**Figure 2 cells-12-01694-f002:**
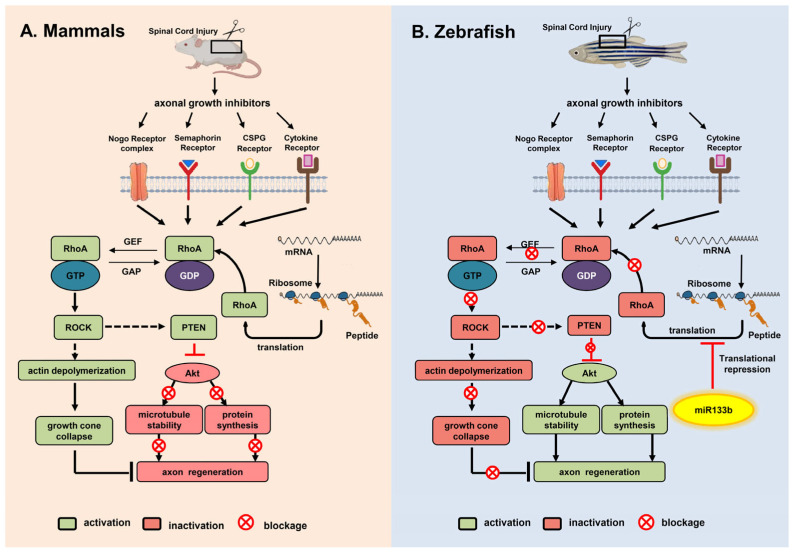
Comparative view of the Rho-ROCK pathway after SCI in mammals and zebrafish and the role of miR-133b in promoting axon regeneration after SCI in zebrafish through the repression of the Rho-ROCK pathway.

**Figure 3 cells-12-01694-f003:**
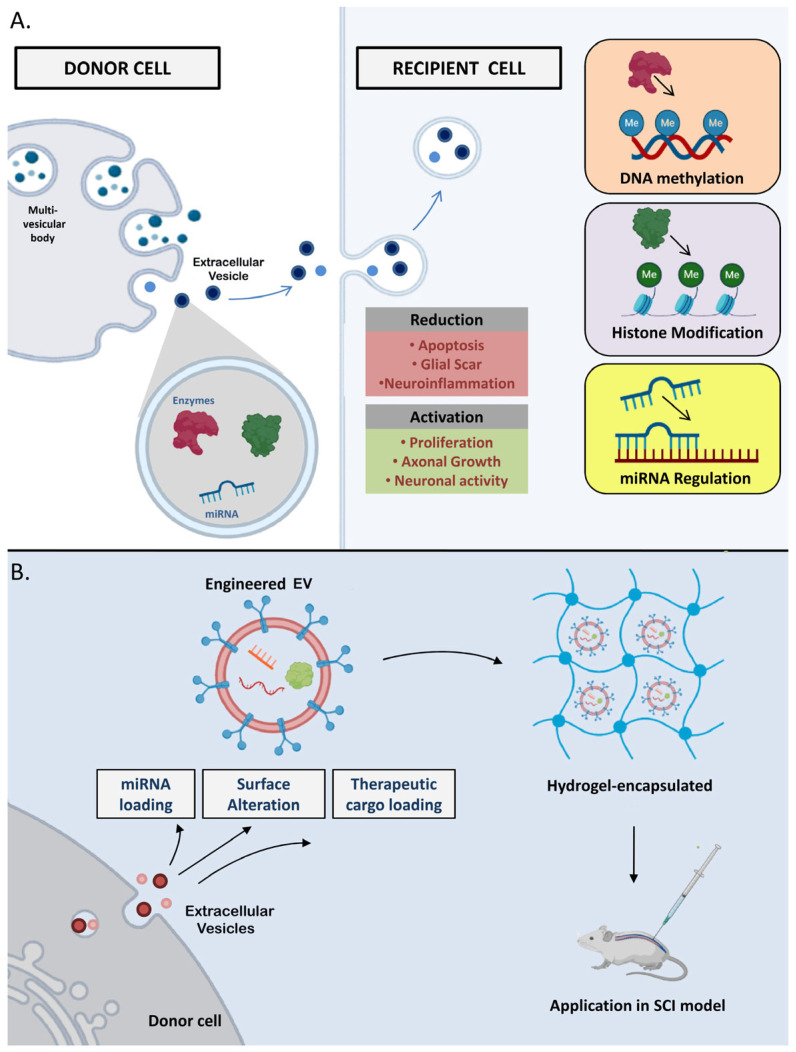
(**A**) Repair mechanisms of the nervous system following SCI after the application of EVs. EVs obtained from donor cells exert diverse epigenetic influences within the microenvironment of the injured spinal cord. These effects involve modifications to DNA, histones, and regulation of noncoding RNA, which collectively contribute to the reduction of glial scar formation and enhancement of axonal growth. (**B**) Strategies for the development of EV-based therapeutics for SCI and evaluation of their regenerative potential. These approaches encompass the design and optimization of EV-based therapies specifically targeted for SCI, including the incorporation of appropriate cargo molecules and modification of EV surface properties. These engineered EVs are then assessed in preclinical SCI models to evaluate their efficacy in promoting CNS-specific regeneration and functional recovery.

**Table 1 cells-12-01694-t001:** List of epigenetic modifications during spinal cord regeneration in regeneration-competent animals.

Different Regenerating Organisms	Types of Epigenetic Modification	Involved Genes/Mediators	Role	Reference
Axolotl	Histone deacetylation	*hdac1*	To control the initial transcriptional response to injury and regeneration in axolotls at the time of tail amputation	[103]
	miRNA regulation	miR-125b	Guides proper axon regeneration involving a direct downstream target, Sema4D, a member of the Semaphorin gene class	[102]
		miR-196	Acts upstream of the main patterning events within the spinal cord based on BMP4 and Pax7	[104]
		miR–200b	Upregulates JunB after SCI, preventing glial cells around the injury site from upregulating the expression of AP-1^cFos/c-Jun^ and inducing glial cells to proliferate	[105]
*Xenopus*	Histone Acetylation	*h3k9ac*	Induces *shh* and *fgf20*	[106]
	Histone Deacetylation	*hdac1*	Induces the expression of two genes, *notch1* and *bmp2*	[107]
		*hdac*	Induces the expression of *mpox* and *spib*, which create an inflammatory response favoring tail regeneration	[108]
Zebrafish	miRNA regulation	miR-133b	Exogenous overexpression of miR-133b increases motor function recovery after SCI by regulating the RhoA signaling pathway	[109]
			Inhibition of miR-133b promotes axon outgrowth via the modulation of *tppp3*	[110]
	Histone Deacetylation	*hdac1*	Induces ependymo-radial glial cell proliferation and increases formation of newborn motor neurons	[111]

## Data Availability

Not applicable.

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
