# Peer review of "Regenerative Potential of Injured Spinal Cord in the Light of Epigenetic Regulation and Modulation"

_cells, 2023, doi:10.3390/cells12131694_

Round 1

Reviewer 1 Report

This is a poorly written review with major problems in assigning of references which are often wrong or out of date. For example, references 1 and 2 in the beginning of the review and 117-122 related to transplantation, which miss the most important and recent work on the subject. In general, the authors lack deep understanding of the complexity of SCI with glaring example is missing any discussion on the autonomic system deficits (bladder, cardiovascular), ignoring the diversity of progenitors (e.g., OPCs), omitting the most important progress on transcription factors that promote regeneration mTOR/PTEN, SOCS3, KLF7 and more, the role of inflammation, and the unsophisticated discussion of the scar.

The introduction and progress report on epigenetics and in particular the work on zebrafish is good reflecting the expertise of the authors, but overall the review is lacking.

editing of sentence and manuscript structure.

Author Response

AUTHOR’S RESPONSE: As pointed out by the reviewer we have revised the following parts in the revised manuscript.

  1. a) When revising our work, we have taken into consideration the valuable feedback provided by the reviewer. To enhance the comprehensiveness and currency of our references, we have incorporated several recent works in addition to the existing citations. By doing so, we aim to provide a more up-to-date and comprehensive overview of the subject matter
  2. b) As pointed out by the reviewer, we have corrected and added an elaborated discussion on glial scar under section 2. Mechanism of injury repair: Glial scar formation
  3. c) As per the reviewer’s suggestion, we have added a paragraph discussing the autonomic system deficits after SCI under section 1. Spinal cord injury: primary and secondary injury in mammals
  4. d) Regarding the mention of the diversity of OPCs and the role of inflammation, we have added a descriptive part under subsection 3.1. Epigenetic regulation of specific cell types and secondary damage processes.
  5. e) In this review, we have not discussed the regulation of regeneration-promoting transcription factors after SCI. However, as suggested, we have added a brief part mentioning the transcription factors that promote regeneration (mTOR/PTEN, SOCS3, KLF7) under section 2. Mechanism of injury repair: Glial scar formation.
  6. f) As pointed out by the reviewer, we have diligently reviewed the entire manuscript and refined the English writing, focusing specifically on enhancing the sentence structures and keeping the science intact.

Reviewer 2 Report

This manuscript examines the epigenetic regulation and role of extracellular vesicles (EVs) in spinal cord injury (SCI) repair and regeneration. It highlights the non-regenerative nature of the human central nervous system and the limited functional recovery in mammalian SCI due to axonal re-development inhibitors and poor intrinsic neuronal reaction. In contrast, certain animals like zebrafish exhibit axon regeneration after SCI. The review focuses on the zebrafish model and discusses epigenetic mechanisms such as DNA methylation, histone modification, and microRNAs that regulate gene expression during repair and regeneration. It emphasizes the conservation of key epigenetic processes between zebrafish and mammals and the potential translational impact on SCI therapies. Additionally, it explores the pathological role of EVs in SCI and their potential therapeutic applications. Overall, the manuscript provides insights into the epigenetic and EV-mediated mechanisms underlying SCI repair and regeneration, aiming to advance the development of treatments for human patients.

The manuscript requires minor revisions.

1.      It is suggested to add subsections related to epigenetic regulation of specific cell types and secondary damage processes in the "Epigenetic regulation during injury repair" section. These subsections should focus on astrocytes and their role in glial formation, oligodendrocytes and the demyelination process, and macrophages (microglia) and their involvement in phagocytosis.

2.      In the "Discussion and therapeutic approaches" section, the manuscript lacks specific recommendations regarding the utilization of extracellular vesicles (EVs) for epigenetic regulation of the regenerative capacity of the mammalian spinal cord. However, it is reasonable to assume that ongoing research is exploring the potential of EVs for this purpose. It is recommended to request further information from the authors regarding any existing or planned research in this area.

3.      To enhance the clarity and completeness of the article, it is suggested to include a table summarizing the identified particles potentially involved in epigenetic regulation and present in extracellular vesicles, along with their functions. Additionally, highlighting relevant studies that analyze the composition of extracellular vesicles would provide valuable information for readers.

4.      More information is requested on the potential implications of epigenetic modulation on spinal cord regeneration for clinical practice.

5.      The manuscript lacks clear discussion on the specific considerations regarding epigenetic regulation of ecogenic progenitors of nerve cells in therapeutic use. Further studies and clarification are necessary to fully understand the mechanisms by which epigenetic regulation plays a role in the therapeutic use of ecogenic progenitors in nerve cell regeneration.

6.      In order to provide a more comprehensive understanding and analysis of the topic at hand, it would be beneficial to include several studies dedicated to extracellular vesicles in the review. These studies, such as DOI 10.3389/fnins.2019.00163,  cover a wide range of topics related to extracellular vesicles. By incorporating these studies, the review can offer a more in-depth exploration of the field of extracellular vesicles and their relevance to the discussed topic.

Author Response

  1. It is suggested to add subsections related to epigenetic regulation of specific cell types and secondary damage processes in the "Epigenetic regulation during injury repair" section. These subsections should focus on astrocytes and their role in glial formation, oligodendrocytes and the demyelination process, and macrophages (microglia) and their involvement in phagocytosis.

AUTHOR’S RESPONSE: As suggested, we have reorganized the Section 2.3. Epigenetic regulation during injury repair. For a better understating of the readers, we have added subsection 2.3.1. Epigenetic regulation of specific cell types and secondary damage processes briefly mentions the astrocytes and their role in the glial formation, oligodendrocytes, and the demyelination process, and macrophages (microglia) and their involvement in phagocytosis as per the reviewer’s requirement.

  1. In the "Discussion and therapeutic approaches" section, the manuscript lacks specific recommendations regarding the utilization of extracellular vesicles (EVs) for epigenetic regulation of the regenerative capacity of the mammalian spinal cord. However, it is reasonable to assume that ongoing research is exploring the potential of EVs for this purpose. It is recommended to request further information from the authors regarding any existing or planned research in this area

AUTHOR’S RESPONSE: We appreciate the reviewer’s suggestion to include specific recommendations regarding the utilization of EVs for epigenetic regulation of the regenerative capacity of the mammalian spinal cord.

While the potential of EVs for epigenetic regulation in the context of spinal cord regeneration is highly promising, the available data on their application in this particular area remains very limited. As highlighted in our manuscript, the current understanding of EV-mediated epigenetic regulation is primarily based on preclinical studies that have demonstrated the transfer of various epigenetic modifiers, such as microRNAs, long non-coding RNAs, and DNA methylation regulators, through EVs.

Some recent studies have indicated modulation of gene expression patterns and cellular processes critical for neuroplasticity and axonal growth. However, it is important to note that the field of EV-based therapies for spinal cord regeneration is still in its early stages, and ongoing research is actively exploring the potential of EVs for this purpose. At present, it is reasonable to anticipate that ongoing studies will shed further light on the efficacy, safety, and optimal utilization of EVs for epigenetic regulation in the regenerative capacity of the mammalian spinal cord. We have briefly mentioned this in the revised manuscript.

  1. To enhance the clarity and completeness of the article, it is suggested to include a table summarizing the identified particles potentially involved in epigenetic regulation and present in extracellular vesicles, along with their functions. Additionally, highlighting relevant studies that analyze the composition of extracellular vesicles would provide valuable information for readers.

AUTHOR’S RESPONSE: We have taken your feedback into consideration and included a table (Table 2.) summarizing the identified biomolecules potentially involved in epigenetic regulation and identified in extracellular vesicles, along with their functions.

We have also incorporated/cited relevant studies that analyzed the composition of EVs carrying these biomolecules.

  1. More information is requested on the potential implications of epigenetic modulation on spinal cord regeneration for clinical practice

AUTHOR’S RESPONSE: In this section 5. Discussion and therapeutic approaches, we wrote about the potential implications of induced pluripotent stem cells (iPSCs) and miRNAs as epigenetic modulators for spinal cord repair. However, we have added a paragraph briefly mentioning the potential implications and reported clinical application of miRNAs as an epigenetic modulator on spinal cord regeneration in section 5 as per the reviewer’s requirement.

  1. The manuscript lacks clear discussion on the specific considerations regarding epigenetic regulation of ecogenic progenitors of nerve cells in therapeutic use. Further studies and clarification are necessary to fully understand the mechanisms by which epigenetic regulation plays a role in the therapeutic use of ecogenic progenitors in nerve cell regeneration.

AUTHOR’S RESPONSE: We appreciate your thoughtful comments regarding the specific considerations related to the epigenetic regulation of ecogenic progenitors in therapeutic nerve cell regeneration. After careful consideration and an extensive search for additional information, we were unable to find sufficient evidence or studies specifically addressing this aspect. As a result, we regret to inform you that we are currently unable to provide a comprehensive discussion on this.

However, assuming mentioning "epigenetic regulation of echogenic progenitors of nerve cells" you refer to the control or modulation of gene expression and cellular function in the progenitor cells that give rise to nerve cells through epigenetic mechanisms, we briefly mentioned this in the revised manuscript under Section 5: Discussion and therapeutic approaches as stated below with appropriate citations.

“In the context of nerve cell regeneration, echogenic progenitors are specialized cells that are frequently introduced or transplanted to a specific location in the body with the aim of generating new nerve cells and facilitating tissue repair. Understanding how epigenetic regulation affects these echogenic progenitor cells can provide insights into their differentiation, maturation, and functional integration into the nervous system. It may also have implications for optimizing their therapeutic use in various neurological conditions and diseases.”

  1. In order to provide a more comprehensive understanding and analysis of the topic at hand, it would be beneficial to include several studies dedicated to extracellular vesicles in the review. These studies, such as DOI 10.3389/fnins.2019.00163, cover a wide range of topics related to extracellular vesicles. By incorporating these studies, the review can offer a more in-depth exploration of the field of extracellular vesicles and their relevance to the discussed topic.

AUTHOR’S RESPONSE: We have now incorporated this study into our revised manuscript, along with several other related studies that cover a wide range of related topics of interest. We believe all these suggestions helped significantly improve the manuscript.

Reviewer 3 Report

This review article "Regenerative potentials of injured spinal cord in the light of epigenetic regulation and modulation", by Gupta S. et al., reviews the modulation of injured spinal cord by epigenetic regulation. The authors reviewed the literature and provided a comparative view of the epigenetic mechanisms between non-regenerative and regenerating species via extracellular vesicles, which has therapeutic potential for intervention.

In this review article, the authors have reviewed and discussed the epigenetic mechanisms in various animal models and the therapeutic potential of extracellular vesicles for the treatment of spinal cord injury. I enjoyed reading this review article.

Author Response

AUTHOR’S RESPONSE: We thank Reviewer for commenting on the manuscript and for valuable input in the assessment.

Round 2

Reviewer 1 Report

Acceptable